# Point-of-care PCR testing of SARS-CoV-2 in the emergency department: Influence on workflow and efficiency

**David Fistera**[1]*, **Katja Kikull**[2], **Joachim Risse**[1], **Anke Herrmann**[3], **Matthias Brachmann**[2], **Clemens Kill**[1]

1 Center of Emergency Medicine, Essen University Hospital, Essen, Germany, 2 bcmed GmbH, Ulm, Germany, 3 Institute for Virology, Essen University Hospital, Essen, Germany

* david.fistera@uk-essen.de

**Data Availability Statement:** All relevant data are within the paper and its Supporting Information files.

## Abstract

### Problem

Regarding transmissible viral diseases such as those caused by SARS-CoV-2 virus, one of the key challenges is isolation management until final diagnosis. This study investigates the influence of SARS-CoV-2 point-of-care (POC) PCR on workflow and efficiency in an emergency department (ED) of a tertiary university hospital.

### Method

An analysis of 17,875 ED patients receiving either SARS-CoV-2 POC PCR (rapid PCR, 11,686 patients) or conventional laboratory SARS-CoV-2 PCR (conventional PCR, 6,189 patients) was performed. The pathways for both groups were mapped and compared, and process times from admission to diagnosis were measured. Effects on resource management within the ED were quantified. Direct costs due to isolation, loss of capacities, and revenues were calculated for inpatients.

### Results

The mean time from admission to result was 1.62 h with rapid PCR and 16.08 h with conventional PCR (p < 0.01), reducing the isolation time by 14.46 h. In the first 2 h after testing, test results were available for > 75% of the rapid PCR group and none of the conventional PCR group. Ninety percent of the results were available within 3 h for the rapid PCR and within 21 h for the conventional PCR group. For the conventional PCR group, an increase in direct costs of €35.74 and lost revenues of €421.06 for each inpatient case was detected.

### Conclusion

Rapid PCR significantly reduces the time-to-results and time for isolation relative to conventional PCR. Although testing costs for rapid PCR are higher, it benefits workflow, reduces total costs, and frees up ward capacity.

**Funding:** The analysis was supported by a grant to Essen University Hospital, Essen, Germany, from Cepheid, Buckinghamshire, UK (www.cepheid.com). None of the authors received a personal grant. The funders had no role in study design, data collection and analysis, decision to publish, or preparation of the manuscript.

**Competing interests:** The authors have declared that no competing interests exist.

## Introduction

The coronavirus disease 2019 (COVID-19) is an infectious disease caused by the SARS-CoV-2 virus (order: Nidovirales, family: Coronaviridae, subfamily: Coronavirinae) [1]. The unpredictability of COVID outbreaks (point of time and incidence rates), the novelty of the virus, and the risk of staff members being infected have put healthcare systems worldwide under unprecedented pressure [2–6]. Uncertainty about a patient's health or infection status upon arrival is a problem for healthcare providers, especially hospitals and their emergency departments (EDs). Clinical COVID diagnoses are difficult because symptoms can resemble other respiratory diseases (e.g., influenza or respiratory syncytial virus [RSV]), or an infected person may be asymptomatic [7,8]. These hurdles make it necessary to isolate all patients; to protect patients and staff members [9]. Isolation management during pandemics can be defined as the key challenge for the ED and inpatient wards.

The standard diagnostic approach for alleged COVID-19 patients is the polymerase chain reaction (PCR) test method [10–12], with test results available within a wide range of up to 24 hours due to varying availability of PCR testing, which can be limited during nights and weekends, and the number of tests. Consequently, alleged COVID-19 patients are fully hospitalized and isolated in a normal ward or in a holding area (blocking a twin bedroom) until the test results are available. Cohort isolation is only possible with a valid test result.

Health authorities of many countries have relied on rapid antigen testing to prevent or reduce the spread of the SARS-CoV-2 virus [13]. Studies on the comparability of the two test methods (antigen and PCR) show that the rapid antigen test is less sensitive, especially among patients with a lower viral load, such as vaccinated patients or those tested very early or late in the course of infection [14–17]. Therefore, PCR testing remains the gold standard for diagnosing SARS-CoV-2 infections.

Point-of-care PCR diagnostic testing of suspected COVID-19 patients can ensure rapid, qualitative, reliable results [18,19]. A point-of-care device has 24/7 availability and can be directly installed in the ED. When point-of-care devices are available, all patients with COVID-19 symptoms can be rapidly diagnosed, discharged, or admitted without isolation or with necessary cohort isolation.

Therefore, this study investigates the influence of point-of-care PCR testing on the workflow and efficiency of a large ED.

## Materials and methods

A process analysis was conducted with the aim of evaluating the operational and economic effects of point of care rapid PCR analysis (rapid PCR) compared to conventional laboratory PCR (conventional PCR) for detecting the SARS-CoV-2 virus in emergency patients from April 1, 2020, to March 31, 2022.

The study was approved by the institutional ethics committee "Ethik-Kommission der Medizinischen Fakultät der Universität Duisburg-Essen" (approval no. 22-10786-BO, date: June 17[th] 2022) and registered in the German trial registry (study no. DRKS00029370).

The study inclusion criteria were: adult patients presenting to the ED during the study period (admission by emergency medical service as well as self-presenting); eventual admission to the hospital (patient status 'inpatient'); valid PCR result of SARS-CoV-2 testing; the time from admission until nasopharyngeal swab for PCR was no longer than 3 hours; the result was received within 24 h after the admission. These criteria were chosen to depict the standard ED process, including its average times (the average length of stay within this specific ED is around 3 h; 3.28 h within German hospitals) [20]. Thus, the effects of possible outliers on the

analysis results were minimized as any deviations from ~~the~~ standard process times were seen as irregularities in the process.

## Data acquisition

Patient data were retrospectively obtained through the electronic medical record (ERPath, eHealth-Tec Innovations GmbH, Berlin, Germany; Medico, Cerner Health Services GmbH, Idstein, Germany). Missing data that could not be extracted from the patients' records were excluded from the statistical analysis.

The study site ED used the Xpert Xpress SARS-CoV-2 test, which is run on Cepheid´s Gen-eXpert. The hospital's laboratory used the Real Star® Kit SARS-CoV2 RT-PCR Kit 1.0 for Altona Diagnostic's Bio-Rad CFX96™ as well as the SARS-CoV-2 AMP Kit for Abbott's Alinity M.

The dataset included timestamps for the admission of the patients, information on which COVID test was performed (conventional PCR or rapid PCR), timestamps for the availability of the test result, and information on relevant additional examinations patients may have needed throughout their stay in the ED (e.g., computed tomography [CT]).

## Process analysis

Fig 1 depicts the two diagnostic pathways for ED patients. There are more steps within the conventional PCR pathway: the sample must leave the ED site and is transported to the virology department, and patients need to be moved to the holding area to wait for their test result in isolation. The uncertainty of the test result makes isolation necessary for a longer period. Hence more process steps need to be completed under isolation. The longer duration of isolation leads to an additional workload for the nursing staff.

Further, this pathway makes disinfection of multiple rooms (ED room, CT suite) necessary, which makes them unusable pending disinfection. Rapid PCR makes several steps obsolete. When reaching certainty about the diagnostic status more quickly, several steps can be carried out without isolation for patients with a negative test result. The latter were the majority in both study groups as described in the results section.

The analysis focused on three major impact factors: the time-to-result, the potentially lost revenues due to isolation, and a direct cost comparison of the two groups. The mean time from admission to test result was calculated from the real-time data for each group. A calculation model was created in line with the two processes to enable the comparison. The cost calculation was performed for the ED process from admission to discharge, focusing on the COVID-specific process steps and comparing the costs between the groups within the time intervals of nasopharyngeal swabs until test results were received. The following cost categories were considered: testing, material, and personnel costs (including medical and non-medical staff).

The consequential costs and revenue information assigned to each relevant process step was based on predefined premises gathered from the ED or virology staff, the University Hospital management (cost and revenue data), and publicly available data or project data from comparable EDs (Table 1). Due to the exceptional circumstances of the pandemic, no variants of the calculation model were created, assuming that the processes did not differ from weekday to weekend-day.

Costs per case were summed up for both study groups showing results for the inpatient category. Calculated results show the total cost and revenue difference per year. Lost revenues were calculated with the German base rates from 2020 through 2022, the case severity indices of the University hospital study site from 2020 through 2022 as the year 2022 was not

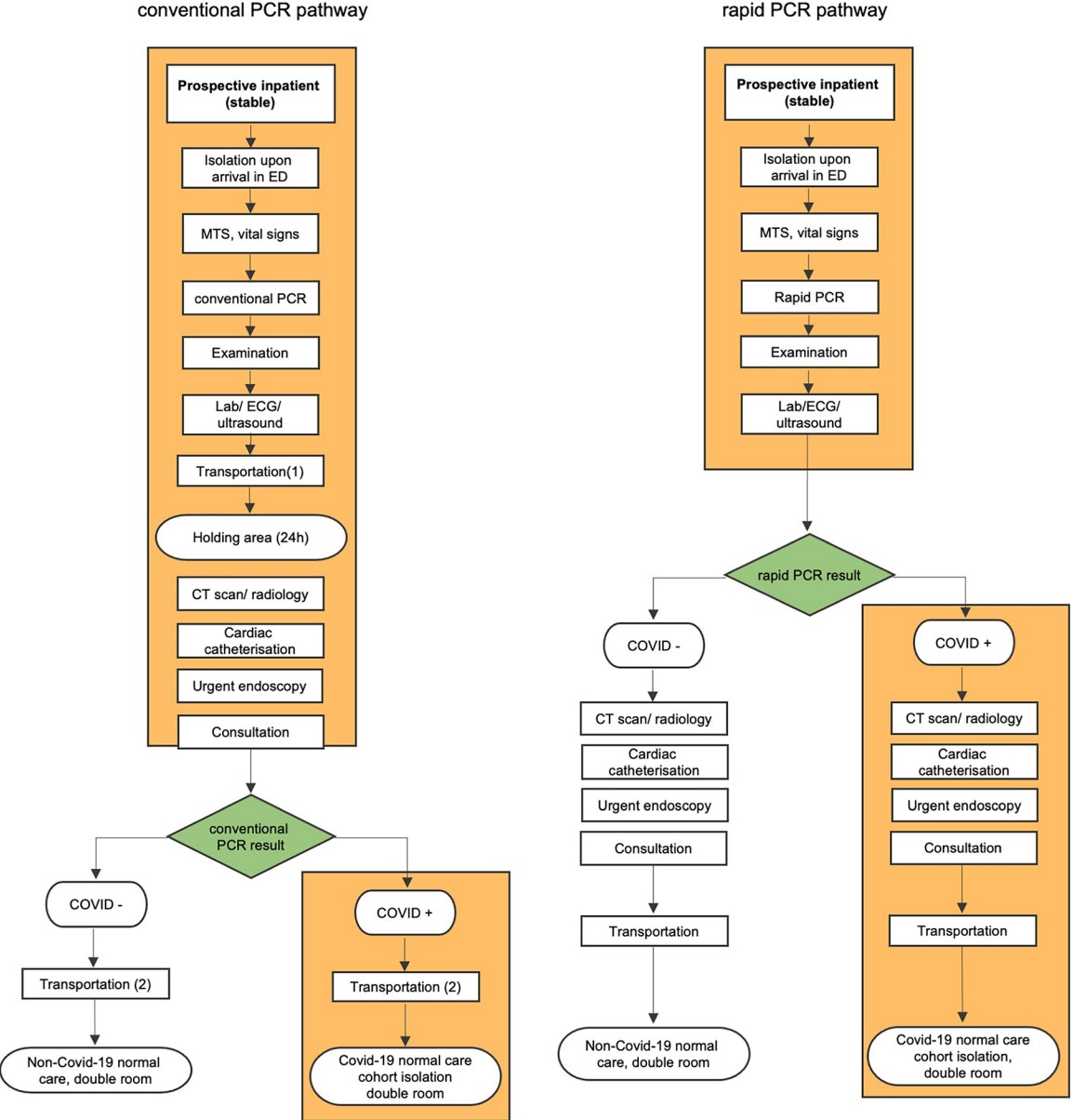

**Fig 1. Two diagnostic pathways for ED patients with conventional PCR vs. rapid PCR.** CT = computed tomography; ED = Emergency Department; MTS = Manchester triage Scale; PCR = Polymerase chain reaction.

completed by the time of the calculation, and an average length of stay in German hospitals of 7.2 days) [21,22]. This calculation is based on standard double ward rooms being most common in German hospitals and assuming that only patients with final diagnosis of infectious status may be cohorted in one room (either both SARS-CoV-2 positive or negative). However, patients with unknown test results have to be isolated (single use of double room).

## Statistical analyses

A t-test was used to evaluate the metric data. To assess the equality of variance, data were tested by Levene's test. Welch's t-test was used to analyze the metric data in cases of unequal

**Table 1. Economic assumptions.**

| position | value |
|---|---:|
| personnel cost: ED nurse p.a.[1] | 60,00 € |
| personnel cost: virologist (physician) p.a. [1] | 96,00 € |
| personnel cost: virology staff p.a. [1] | 40,00 € |
| personnel cost: radiographer p.a. [1] | 50,00 € |
| personnel cost: cleaning staff p.a. [1] | 36,00 € |
| lab PCR testing cost [2] | 11.68 € |
| rapid PCR testing cost[3] | 35.20 € |
| German nationwide base rate (average 2020–2022) [21] | 3,72 € |
| average length of stay in days (2021) [22] | 7.2 |

[1] bcmed project data.

[2] information from the study site's virology (calculated as an average of two conventional PCRs used with utilization rates from the study population).

[3] market price Cepheid SARS-CoV-2 cartridge (fall 2022).

[21] National Association of Statutory Health Insurance Funds.

[22] German Federal Statistical Office.

variances. Results were reported as mean ± standard deviation for metric variables. Pearson's chi-square or Fisher's exact test were used to evaluate the categorical data. Results for the categorical variables were reported as percentages. Statistical significance was defined as two-tailed $p < 0.05$. Data were processed using Excel software (Version 2017, Microsoft Corporation, Redmond, WA, USA) and IBM SPSS Statistics (Version 29.0, IBM Corporation, Armonk, NY, USA).

## Results

### Study population

A total of 17,875 patients were included into the analysis (6,189 conventional PCR, 11,686 rapid CPR). Of those patients, 9,678 (54.1%) were prospective inpatients. The mean age was 52.4 (± 21) years in the conventional PCR group and 60.3 (± 20) years in the rapid PCR group. There were 6,189 patients in the conventional PCR group and 11,686 patients in the rapid PCR group, 2,990 (48.3%) and 6,257 (53.6%) were male, respectively. The positive rate fort the whole study population was 10.7%, while for all inpatients it was 9.9%.

Baseline characteristics as well as outcome parameters of inpatients are summarized in Table 2

### Time-to-test-result

Mean time from admission to test result was 97 (± 48) minutes in the rapid PCR group and 965 (±339) minutes in the conventional PCR group (p<0.001). This equals 1.62 h for the rapid PCR and 16.08 h for the conventional PCR. Rapid PCR was 867 minutes (14.46 h) faster to produce reliable results than conventional PCR.

While 100% is already reached within 3 h with rapid PCR, the conventional PCR requires 21–24 h (Fig 2). The 1.5–2-h time window is also noteworthy: while about 75% of the patients had already received their test results using rapid PCR, conventional PCR had not yet delivered a single result. For conventional PCR, it takes 18–21 hours to achieve 75% of the test results delivered.

**Table 2. Statistical analysis (group comparison).**

| parameter | | | rapid PCR N = 11,686 | conventional PCR N = 6189 | P value |
|---|---|---|---|---|---|
| Age [years] | [years] | Mean (±SD) | 60.3 (±19.9) | 52.4 (±21.0) | |
| Gender male | | N (%) | 6,257 (53.5) | 2,990 (48.3) | <0.001 |
| Inpatients | | N (%) | 7,589 (64.9) | 2,089 (33.8) | <0.001 |
| COVID positive | | N (%) | 995 (8.5) | 823 (13.4) | <0.001 |
| Time to result | [minutes] | Mean (±SD) | 97 (±48) | 965 (±339) | <0.001 |
| MTS category | Red | N (%) | 2,460 (21.1) | 196 (3.2) | <0.001 |
| | Orange | N (%) | 1,888 (16.2) | 572 (9.2) | <0.001 |
| | Yellow | N (%) | 3,402 (29.1) | 1,812 (29.3) | <0.001 |
| | Green | N (%) | 3,512 (30.1) | 2,577 (41.6) | <0.001 |
| | Blue | N (%) | 395 (3.4) | 1,017(16.4) | <0.001 |
| | N/A | N (%) | 29 (0.2) | 15 (0.2) | |
| Arrival mode | Self | N (%) | 3,480 (29.8) | 2,775 (44.8) | <0.001 |
| | Ambulance | N (%) | 6,406 (54.8) | 2,651 (42.8) | <0.001 |
| | Referral | N (%) | 1,218 (10.4) | 404 (6.5) | <0.001 |
| | N/A | N (%) | 582 (5.0) | 359 (5.8) | |
| CT/ MRI | | N (%) | 4627 (39.6) | 1446 (23.4) | <0.001 |
| Chest X-ray | | N (%) | 3012 (25.8) | 839 (13.6) | <0.001 |
| Cardiac cath. | | N (%) | 450 (3.9) | 56 (0.9) | <0.001 |
| Endoscopy | | N (%) | 483 (4.1) | 61 (1.0) | <0.001 |

**Outcome (inpatients only)**

| | | | rapid PCR N = 7589 | conventional PCR N = 2089 | p-value |
|---|---|---|---|---|---|
| IMC/ICU | | N (%) | 2,175 (28.7) | 393 (18.8) | <0.001 |
| Ventilator | | N (%) | 1,150 (15.2) | 197 (9.4) | <0.001 |
| mortality | | N (%) | 956 (12.6) | 175 (8.4) | <0.001 |
| Length of stay | [days] | Mean (±SD) | 5.9 (±10.6) | 2.7 (±6,7) | <0.001 |

## Lost revenues

Considering the disinfection and isolation time, the blocked ward capacity for one inpatient adds up to 15.06 h (15 h and 3 min), translating into €421.06 of lost revenues for one inpatient with a negative test result: the base rates of Germany hospitals for 2020 through 2022 were multiplied with the hospitals case severity indices of the respective year leading to €4,830.68 average revenue per case. This was adapted to 15.06 hours of blocked ward capacity leading to 421.06€ of lost revenues.

Extrapolating to one pandemic year led to 65,604 h of blocked ward capacity. These could have been utilized for a calculated 380 other patients with an anticipated total reimbursement of €1,833,971.11 over one year (Table 3).

## Direct costs

The cost difference between the two pathways for one inpatient with a negative test result was €35.74 in favor of the rapid PCR (Table 4); the conventional PCR costs were €91.41, and the rapid PCR cost was €55.67.

In the conventional PCR pathway, the main cost driver was the other personnel costs, which amounted to €47.97 (Fig 3). These costs include expenses for the transport service,

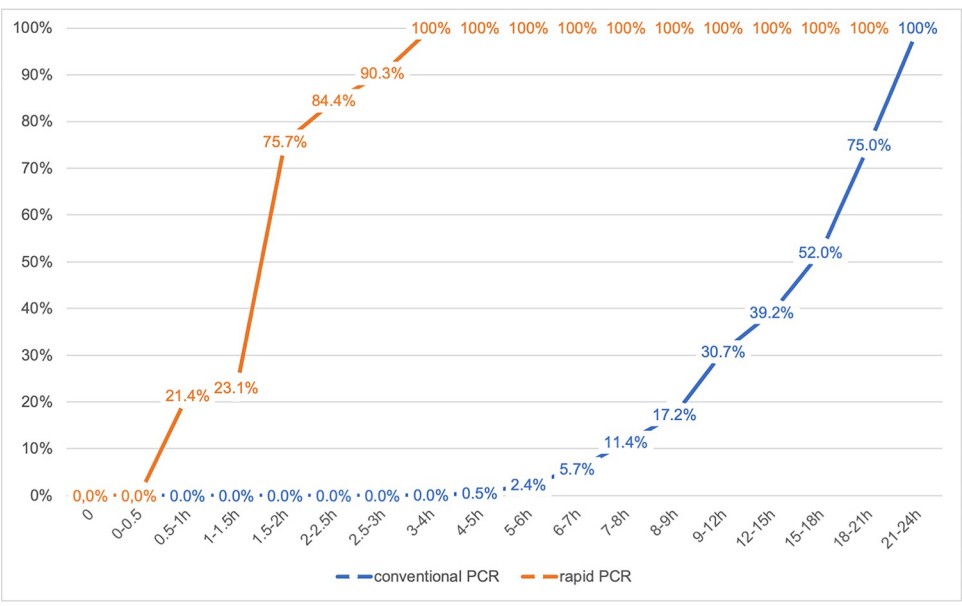

**Fig 2. Cumulated percentage of test results in specific periods.**

laboratory staff, cleaning personnel, and radiographer waiting for the CT suite to be disinfected. On the other hand, the rapid PCR pathway has testing costs of €35.20, which account for more than half of the total costs and are three times higher than the testing costs for the conventional PCR pathway. However, the nursing staff costs differ by €11.93 between the two pathways, with the rapid PCR pathway being less expensive.

Extrapolating these findings to the study population per year, the cost difference was estimated to be €155,652.81 for inpatients with a negative test result (Table 5).

## Discussion

During the past years of the COVID pandemic, isolation management until a final diagnosis has proven to be one of the key challenges for EDs. This analysis was designed as a retrospective study using real patient data, including the initial period of the pandemic in March 2020, with its uncertainties, overcrowding situations, and interim process improvements over the course of the 2-year pandemic. Thus, it provides a valid data basis for real-life conditions in a 24/7 ED.

**Table 3. Blocked ward capacity and potential revenues.**

|  | one case | study population |
|---|---|---|
| I̶inpatients per year* |  | 4,839 |
| Inpatients per year with negative test result |  | 4,356 |
| Difference in time to test result (= isolation time) in h | 14.46 | 62,990 |
| Time for disinfection (conventional PCR only) in h | 0.6 | 2,613 |
| **time saved (h) (patients*difference)** | **15.06** | **65,604** |
| **potential revenues** | **412.06 €** | **1,833,971.11 €** |

*17.875 patients / 2 years X 54,1% inpatient-rate.

**Table 4. Direct cost comparison of the two diagnostic pathways for one inpatient case.**

| Process step | Cost position | Conventional PCR | Rapid PCR | Difference |
|---|---|---|---|---|
| Diagnostic test | | | | |
| | Cost of the test | 11.68 € | 35.20 € | -23.52 € |
| | Cost of nurse handling rapid test / preparing sample | 1.33 € | 3.32 € | -1.99 € |
| | Transport of sample to virology | 5.00 € | - € | 5.00 € |
| | Cost of staff handling and approving conventional test | 7.25 € | - € | 7.25 € |
| Patient isolation in ED | | | | |
| | Disinfection of ED room patient is leaving | 14.32 € | - € | 14.32 € |
| | Transport of patient to holding area | 9.95 € | - € | 9.95 € |
| | Add. personnel cost for isolation measures in holding area | 3.98 € | - € | 3.98 € |
| | protective clothing | 11.50 € | 0.54 € | 10.96 € |
| Possible CT | | | | |
| | Disinfection of CT unit* | 2.96 € | 0.96 € | 2.00 € |
| | Lost radiographer time | 4.11 € | 1.33 € | 2.78 € |
| | Protective clothing | 1.66 € | - € | 1.66 € |
| Physicians consultation | | | | |
| | Protective clothing | 3.35 € | - € | 3.35 € |
| Patient admission | | | | |
| | Cleaning of last patient room | 14.32 € | 14.32 € | - € |
| **Total costs** | | **91.41 €** | **55.67 €** | **35.74 €** |

*only 49,5% of inpatients receive a CT scan, disinfection takes approx. 15 mins.

The analysis compared the procedural and cost effects of two different PCR diagnostic pathways in a university hospital ED. The results demonstrate that using a POC PCR test in the context of the COVID-19 pandemic significantly streamlines processes and helps ensure work capacity and throughput.

There are significant positive effects on time-to-result, room capacity in terms of potential revenues, and direct costs for inpatients. The operational advantages of rapid PCR over conventional PCR are best seen in the reduction of the time-to-result: less isolation time is needed, less workload is produced when isolation time is reduced (especially for the nursing staff), there are fewer process steps and interfaces (Fig 1), everything is kept in one place, and a single nurse can manage the complete process from nasopharyngeal swab to test result. These effects subsequently affect the direct costs and the used ward capacity.

Considering baseline characteristics as well as additional intrahospital diagnostic tests and outcome parameters, both groups revealed to be significantly different. These differences can best be explained by the evolution of POCT PCR testing during the pandemic. In the beginning, availability of testing cartridges was highly limited and expensive, so unstable and severely ill patients received rapid PCR testing to ensure a fast and safe transfer to the IMC and ICU. Consequently, the rapid PCR group showed a higher MTS rating, significantly higher numbers of CT-scans, IMC/ICU admissions, and a higher overall mortality. Most outpatients had been tested by conventional PCR at that stage of the pandemic, leading to a lower MTS rating and a higher number of outpatients among the conventional group. As POCT cartridges became broadly available over time, virtually all patients were tested on scene.

These significant differences, however, do not affect the general results and findings of this study: all calculations are based on real life data and a higher rate of additional testing in the rapid PCR group would reduce the cost difference between groups.

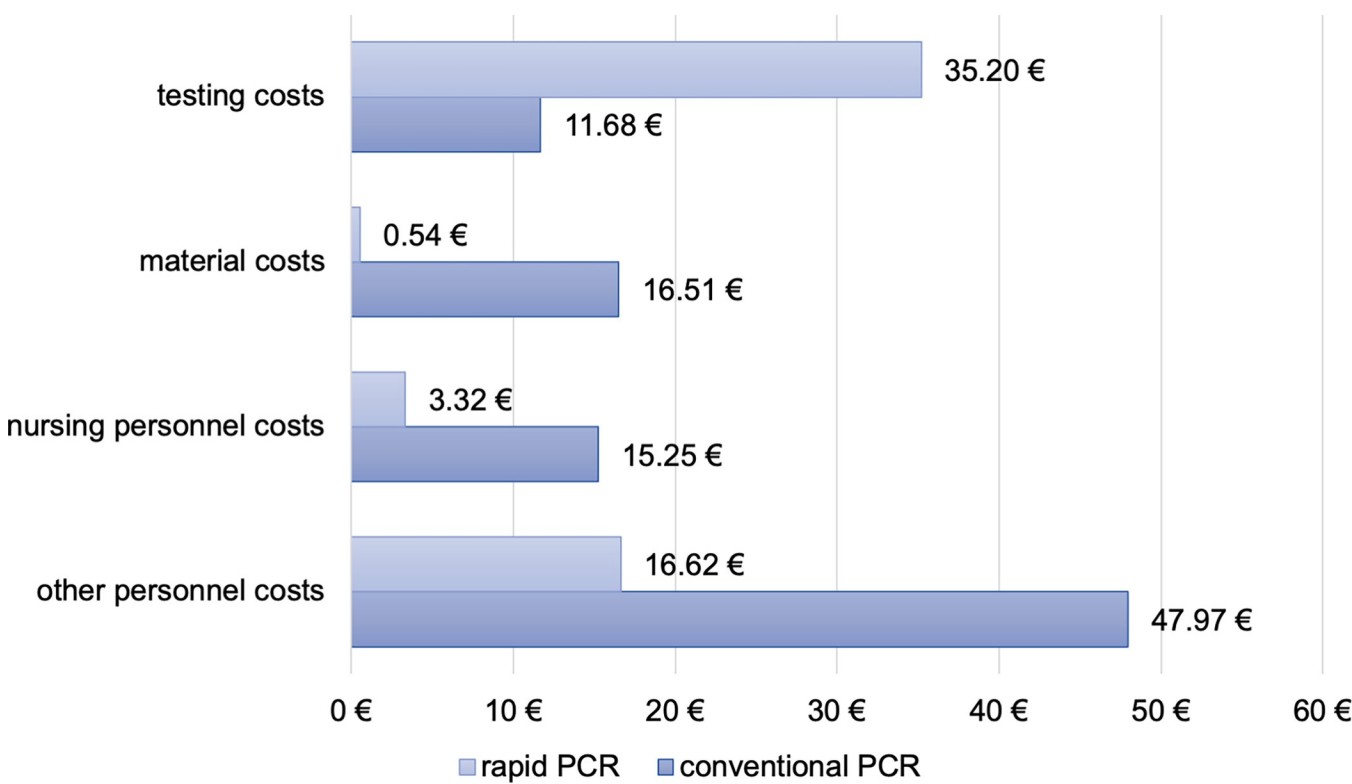

**Fig 3. Cost comparison of one inpatient in different cost categories.**

Furthermore, the assumptions of our calculations have been chosen cautiously. The conventional pathway represents a best-case scenario in terms of laboratory accessibility, as the study site is a university hospital ED with a laboratory on campus and 24-hour access. Smaller hospitals may have more difficult preconditions, with off-site laboratories resulting in longer transport service times and longer waiting times for conventional PCR results due to the laboratorys´ working hours.

The worldwide pandemic situation can and will eventually come to an end [23]. However, the virus itself cannot be eliminated and it is uncertain whether and when the next virus diseases will spread. Therefore, assessment of the economic and procedural effects of using a rapid PCR test is crucial to evaluate future management within healthcare facilities and provide insights into a reliable rapid diagnostic pathway during pandemics or other crowding situations. The rapid PCR test might be feasible for future new viral mutations of the SARS-CoV-

**Table 5. Total costs for the two diagnostic pathways.**

|  | One case | Study population |
| --- | --- | --- |
| Inpatients per year* |  | 4,839 |
| Inpatients per year with negative test result |  | 4,356 |
| Conventional PCR pathway | 91.41 € | 398,127.86 € |
| Rapid PCR pathway | 55.67 € | 242,475.05 € |
| **Cost difference** | **35.74 €** | **155,652.81 €** |

*17.875 / 2 years * 54% inpatientrate.

2 virus and other respiratory pathogens requiring isolation, such as flu and RSV. It can be used in the ED to free up room and personnel resources, resulting in lower expenses and more resources for additional revenue.

## Limitations

The analysis does not provide a complete process cost calculation but focuses on the cost differences between the two pathways. Regarding the cost analysis, it is important to note that the organizational effects for the hospital of searching for an inpatient bed and the associated personnel costs, and the possibility of canceling planned procedures were not considered. In addition, the reputation effects due to longer waiting times for other patients were not included since they were difficult to calculate realistically.

Furthermore, the study focused solely on the COVID virus, and other respiratory diseases in the ED may also require patients to be isolated upon admission to the hospital. This means that even though most patients received a negative COVID test result, some would still have to be kept under isolation with the suspicion of other infectious respiratory diseases, leading to prolonged isolation times. However, during the COVID pandemic, the rates of influenza and RSV in Germany were extremely low, so this effect seems negligible for the calculations.

This study was not intended to compare patient groups, but the differences in workflow between two distinct different testing strategies. Therefore, the significant differences between groups need to be mentioned but did not affect our assumptions and calculations.

Despite these limitations, this study provides evidence that rapid PCR testing in the context of respiratory pathogens requiring isolation benefits workflow, reduces total costs, and frees up ward capacity. The findings of this study have important implications for healthcare providers and policymakers in their efforts to manage future challenges like the COVID-19 pandemic.

## Conclusion

Rapid PCR significantly reduces the time-to-result and time for isolation relative to conventional PCR. Although testing costs for rapid PCR are higher, it benefits workflow, reduces total costs, and frees up ward capacity. Rapid PCR may therefore be beneficial in other respiratory infections such as RSV and influenza and may help to cope with future pandemic challenges.

## Supporting information

**S1 Checklist. STROBE statement—checklist of items that should be included in reports of observational studies.**
(PDF)

**S1 Dataset.**
(XLSX)

## Acknowledgments

We acknowledge Henrik Braitsch for the preparation of the dataset for our study as well as Ute von Frantzki and Nico Päuler as heads of the ED nursing team who contributed with important insights into the specifics to the two pathways.

## Author Contributions

**Conceptualization:** David Fistera, Joachim Risse, Matthias Brachmann.

**Data curation:** Joachim Risse.

**Formal analysis:** Katja Kikull, Joachim Risse.

**Funding acquisition:** Clemens Kill.

**Investigation:** David Fistera, Anke Herrmann.

**Methodology:** Anke Herrmann.

**Project administration:** Katja Kikull, Matthias Brachmann.

**Resources:** David Fistera, Anke Herrmann, Matthias Brachmann.

**Software:** Joachim Risse, Matthias Brachmann.

**Supervision:** David Fistera, Clemens Kill.

**Validation:** Joachim Risse, Clemens Kill.

**Visualization:** Katja Kikull.

**Writing – original draft:** David Fistera, Katja Kikull.

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
