## [Decision Letter · Decision Letter 0]

1 Jun 2023

PONE-D-23-14019Point-of-care PCR testing of SARS-CoV-2 in the emergency department: Influence on workflow and efficiencyPLOS ONE

Dear Dr. Fistera,

Thank you for submitting your manuscript to PLOS ONE. After careful consideration, we feel that it has merit but does not fully meet PLOS ONE’s publication criteria as it currently stands. Therefore, we invite you to submit a revised version of the manuscript that addresses the points raised during the review process.

We look forward to receiving your revised manuscript.

Kind regards,

Vittorio Sambri, M.D., Ph.D.

Academic Editor

PLOS ONE

Journal Requirements:

   "funding statement: The analysis was supported by a grant to Essen University Hospital, Essen, Germany, from Cepheid, Buckinghamshire, UK."

Additional Editor Comments:

Your paper has been evaluated by two experts in the field of COVID-19 diagnostic and Health Care organization: both suggested that a major review  must be undertaken in order to improve the value of your work.

Please see below for the detailed indications by the reviewers.

In summary, I  suggest that you:

1. include a more detailed analysis (see reviewer n.1) about the overall difference among the two populations;

2. discuss, maybe by including a "prospective  analysis" of the "true daily life" impact of POCTs like testinf (as suggested by reviewer n.2);

3. include more details about the clinical picture of the studied patients, since this could help to better understand the impact of POCT testing on each individual patient management.

Reviewers' comments:

Reviewer's Responses to Questions

**Comments to the Author**

1. Is the manuscript technically sound, and do the data support the conclusions?

Reviewer #1: Yes

Reviewer #2: Partly

2. Has the statistical analysis been performed appropriately and rigorously? 

Reviewer #1: Yes

Reviewer #2: N/A

3. Have the authors made all data underlying the findings in their manuscript fully available?

Reviewer #1: No

Reviewer #2: No

4. Is the manuscript presented in an intelligible fashion and written in standard English?

Reviewer #1: No

Reviewer #2: Yes

5. Review Comments to the Author

Reviewer #1: In general:

Your manuscript addresses and important aspect of Point-of-care diagnostics for infectious disease, which is the economical impact (benefit / cost). Your documentation is convincing, but a thorough description of the differences between the two populations (rapid vs standard PCR) is lacking. E.g. were there any differences in use of additional diagnostics, such as CT-scans etc? As you will agree, contact isolation may (will likely) have adverse impact on patient management, such as fewer /postponed diagnostics (e.g. CT-scan), fewer physical contacts (e.g. EWS by nurse).

Specific:

Page 3, line 60: “COVID diagnoses” – do you mean clinical COVID diagnoses?

Page 3, line 63: “protect them” – I guess you mean to protect non-infected patients?

Page 3, line 69: “twin bedroom” – you mean any rooms with capacity >1 patient?

Page 3, line 70: “Cohort isolation is only possible with a valid test result” Please be more specific. As you correctly state in line 60, clinical symptoms may not differentiate SARS-CoV-2 infection from infections by Influenza or RSV – or other respiratory infections. Surely you do not mean that a negative SARS-CoV2 result facilitates cohort isolation of all patients with respiratory symptoms?

Page 4, line 85: Please describe the criteria that was applied for selection of patients for rapid PCR

Page 4, line 96: Why is results received after 24 hours an exclusion criterion?

Page 5, line 100: “…were seen as irregularities…”. I disagree. A result later than 24 hours may reflect problems in the lab, e.g. staffing, shortage of supplies/kits etc. This should be taken into account, otherwise you may underestimate the impact of rapid (PoC) diagnostics.

Page 5, line 111: “…administration of the patient…”. Do you mean “admission”?

Page 5, line 112: Did you include the timestamp for sampling of the patient?

Page 7, line 149, Table 1:

- What does “p.a.” mean?

- Is the average lab cost for PCR only EUR12? I believe this to be true for in-house Altona assays, bit what was the cost for Abbott Alinity tests?

Page 8, line 178: What was the split between inhouse and Alinity of the 6,189 conventional PCR tests?

Page 8, line 182: “..rate of CT-scans performed was…”. This reflects back to my general question above about impact on patient management. Was the lower rate of CT scans in the conventional PCR group correlated to less severe clinical presentation? I.e. was the lower rate clinically justified?

Page 10, line 218: Cost of test in conventional PCR is only EUR11.68. Again, I believe this to be true for in-house Altona assays, bit what was the cost for Abbott Alinity tests.

Page 12, line 254: Is this your best explanation? Do you have any data on the lower rate of CT scans in the conventional PCR group was correlated to less severe clinical presentation?

Reviewer #2: In my opinion there are two main aspects that need to be pointed out about the study.

First, due to the rectrospective nature the authors calculate the costs and the benefit on workflows ideally. The authors do not take in consideration all the other factors that can intervene in real life management of ED patients, that creates the necessity to mantain isolation or contact precautions, like other respiratory communicable disease or colonization by MDR bacteria.

Moreover they do not describe the main clinical characteristics of patients population, which can influence the time needed to diagnose and manage every single case in ED, hence modyfing costs and workflows. Briefly the study, even if designed on correct assumption, oversimplify real life patients management in ED and relative costs and time.

Second, at this time POCT PCR test for Sars Cov2 are largely used in ED and their advantages are well testified by real life, so, in my opinion the present study results arrives late and add little to scientific knowledge.

6. PLOS authors have the option to publish the peer review history of their article (what does this mean?). If published, this will include your full peer review and any attached files.

Reviewer #1: No

Reviewer #2: No

---

## [Author Response · Author response to Decision Letter 0]

1 Jul 2023

Dear Reviewer #1,

Thank you for the thorough review of our manuscript, the plausible criticism, and your overall professional opinion. We are thankful for the opportunity to revise our manuscript according to the points raised and are pleased to present you with the updated version of the manuscript. Please find our explanations regarding the points made below in blue.

Reviewer #1: In general:

Your manuscript addresses and important aspect of Point-of-care diagnostics for infectious disease, which is the economical impact (benefit / cost). Your documentation is convincing, but a thorough description of the differences between the two populations (rapid vs standard PCR) is lacking. E.g. were there any differences in use of additional diagnostics, such as CT-scans etc? As you will agree, contact isolation may (will likely) have adverse impact on patient management, such as fewer /postponed diagnostics (e.g. CT-scan), fewer physical contacts (e.g. EWS by nurse).

Thank you for raising this concern. We included a table with baseline characteristics, early diagnostic procedures, and outcome data for both groups to enhance transparency. Differences between groups are obvious and significant but can be attributed to the evolution of POCT testing in our ED. Since we started by POCT testing the seriously ill/shock room patients, the rapid PCR group shows a higher rate of emergency service admissions, a higher MTS rating, and a higher rate of additional testing (x-ray/ct-scans/endoscopy/cardiac catheterization). However, all assumptions and calculations were not biased by this since the time spans were measured under real-life conditions and the pathways remained the same irrespective of the patients’ health state or preconditions. 

Specific:

Page 3, line 60: “COVID diagnoses” – do you mean clinical COVID diagnoses?

Yes, that is what we mean. Thank you, we corrected this in the manuscript.

Page 3, line 63: “protect them” – I guess you mean to protect non-infected patients? 

We meant all patients and specified this in the manuscript. We apologize for the confusion.

Page 3, line 69: “twin bedroom” – you mean any rooms with capacity >1 patient? 

A “twin bedroom” means a two bedroom, which is a standard hospital inpatient room in Germany. At the study site hospital there are only inpatient rooms with two beds in them. 

Page 3, line 70: “Cohort isolation is only possible with a valid test result” Please be more specific. As you correctly state in line 60, clinical symptoms may not differentiate SARS-CoV-2 infection from infections by Influenza or RSV – or other respiratory infections. Surely you do not mean that a negative SARS-CoV2 result facilitates cohort isolation of all patients with respiratory symptoms? 

We agree that there are more respiratory infections that lead to mandatory isolation. However, due to the extremely low case numbers of influenza and RSV during the COVID-19 pandemic, we felt that these could be neglected for our calculation. Since September 2022 we switched to a fourfold test kit including Influenza A+B, RSV, and COVID for all patients suspected of having respiratory diseases. 

Page 4, line 85: Please describe the criteria that was applied for selection of patients for rapid PCR

In the beginning, medical urgency determined which patient received which PCR test, meaning that severely ill or unstable patients preferentially received rapid PCR testing. Over the course of the pandemic, more and more patients were tested with the rapid PCR test, depending on a greater availability of the test kits. In the end, all emergency patients received rapid PCR testing.

Page 4, line 96: Why is results received after 24 hours an exclusion criterion? 

When the Delta variant of SARS-CoV-2 virus occurred in Germany, additional variant analysis was undertaken in the conventional viral lab with results often becoming available after more than 24h. To exclude this bias, we chose to exclude those “late” results after discussing the case with our colleagues from the department of virology, since they do not represent the pure testing.

Page 5, line 100: “…were seen as irregularities…”. I disagree. A result later than 24 hours may reflect problems in the lab, e.g. staffing, shortage of supplies/kits etc. This should be taken into account, otherwise you may underestimate the impact of rapid (PoC) diagnostics. 

See comment above. 

Page 5, line 111: “…administration of the patient…”. Do you mean “admission”? 

We apologize for the confusion. “Administration” means the process of creating a new patient case in the hospital information system. Since “administration” always takes place at admission to the ED, we replaced that term with “admission.”

Page 5, line 112: Did you include the timestamp for sampling of the patient? 

No, this timestamp was not available for the rapid PCR test. We therefore used the timestamp “admission” to have a real-life timespan for both groups from admission to test result. We feel that this is the safest and most reliable way available. From real life experience we want to share that there are only a few minutes in between the admission of a patient and the collection of a sample since all ED employees were trained to take specimens first.

Page 7, line 149, Table 1:

- What does “p.a.” mean? 

This means “per annum”.

- Is the average lab cost for PCR only EUR12? I believe this to be true for in-house Altona assays, bit what was the cost for Abbott Alinity tests? 

Yes, you are correct that The Abbott Alinity test is costlier. As stated in the footnotes below Table 1 in the manuscript the 12 Euros were calculated as an average of two costs for two different conventional PCRs with the utilization rates calculated from the study. However, the 12 Euros are rounded, and the calculation (please see below) sums up to 11.68 Euros. We corrected this in the manuscript.

The calculation of the 11.68 is as follows: 

Cost for CFX Bio Rad PCR: 7.25 Euros used in 55% of the 6,189 cases in the conventional lab study group

Cost for Alinity Abbott PCR: 17 Euros used in 45% of the 6,189 cases in the conventional lab study group

7.35 Euros X 55% + 17 Euros X 45% = 11.68 Euros 

Page 8, line 178: What was the split between inhouse and Alinity of the 6,189 conventional PCR tests? 

CFX Bio Rad PCR: 55% and Alinity Abbott PCR: 45% 

Page 8, line 182: “..rate of CT-scans performed was…”. This reflects back to my general question above about impact on patient management. Was the lower rate of CT scans in the conventional PCR group correlated to less severe clinical presentation? I.e. was the lower rate clinically justified? 

Yes, you are right. Table 2 provides additional clinical data of both groups. According to a lower MTS rating, a higher outpatient rate, and a lower rate of ICU/IMC treatment, the conventional PCR group was less severely ill. Thus, the lower rate of additional diagnostics was clinically plausible. The reason for the lower rates has been discussed above (“general question”). 

Page 10, line 218: Cost of test in conventional PCR is only EUR11.68. Again, I believe this to be true for in-house Altona assays, bit what was the cost for Abbott Alinity tests. 

Yes, this is correct. As stated above (note to Page 7, line 149, Table 1), the 12 Euros were calculated as an average of the costs for two different conventional PCRs with the utilization rates calculated from the study. However, the 12 Euros are rounded, and the calculation sums up to 11.68 Euros. We corrected this in the manuscript.

Page 12, line 254: Is this your best explanation? Do you have any data on the lower rate of CT scans in the conventional PCR group was correlated to less severe clinical presentation? 

Yes, we presented additional clinical data in the new Table 2 and discussed it in the section “Discussion”. The link between the lower rate of additional diagnostics such as CT scans and less severe case presentation has now been established.

 

Dear Reviewer #2,

Thank you for your overall professional opinion, for raising critical questions and helping us improve our work. We are thankful for the opportunity to revise our manuscript according to the points raised and are pleased to present you with the updated version of the manuscript. Please find our explanations regarding the points made below in blue.

Reviewer #2: In my opinion there are two main aspects that need to be pointed out about the study.

First, due to the retrospective nature the authors calculate the costs and the benefit on workflows ideally. The authors do not take in consideration all the other factors that can intervene in real life management of ED patients, that creates the necessity to maintain isolation or contact precautions, like other respiratory communicable disease or colonization by MDR bacteria.

You are right that there are more respiratory infections that lead to mandatory isolation. However, due to the extremely low case numbers of alternative pathogens requiring isolation, we felt that these could be neglected for our calculation during the COVID pandemic. RSV and influenza virus were rarely detected during the COVID-19 pandemic, probably due to the widespread use of face masks and social distancing. From September 2022 the study site ED switched to the fourfold test kit including Influenza A+B, RSV, and COVID. 

We agree that the retrospective design of our study is prone to simplifications and “ideal calculations”. To minimize these effects, we chose to calculate the timespans from available and reliable timestamps in the clinical information system. We believe that the mean time to result in the rapid PCR group (97 ± 48 minutes) is a far more realistic one than the pure 45 minutes the manufacturer claims for testing.

Moreover they do not describe the main clinical characteristics of patients population, which can influence the time needed to diagnose and manage every single case in ED, hence modyfing costs and workflows. Briefly the study, even if designed on correct assumption, oversimplify real life patients management in ED and relative costs and time. 

To enhance transparency, we added relevant clinical data on both patient groups in the results section (Table 2). Although these data show obvious and significant differences between both groups, we want to emphasize that a) differences can consistently be explained by the way of introducing rapid PCR in our ED and b) they do not affect the general assumptions and calculations. Irrespective of the patients´ case severity, the conventional and the rapid PCR testing pathways were carried out in the same steps. We hope that these data will not cause further confusion since they suggest that we compared heterogenous groups. This is in fact true, but a clinical group comparison was not intended.

Second, at this time POCT PCR test for Sars Cov2 are largely used in ED and their advantages are well testified by real life, so, in my opinion the present study results arrives late and add little to scientific knowledge. 

According to our observations, the POCT PCR test is not yet established in many EDs since they are believed to be “too expensive” and the real cost of not testing rapidly is underestimated. Therefore, a solid calculation including all relevant costs, especially caused by longer times of isolation and blocked ward capacities, is highly relevant in our eyes. 

In November and December 2022, Germany was suffering from a relevant RSV wave in the pediatric emergency departments but also the adult ED. At the same time, influenza reached an unusual early peak, so reliable and fast testing remains a relevant topic in future winter seasons and our results can be generalized for those two common viral diseases. 

To our knowledge there are few comparable data sets and studies focusing on a well-set calculation of costs and efficiency.

---

## [Editor Report · Decision Letter 1]

6 Jul 2023

Point-of-care PCR testing of SARS-CoV-2 in the emergency department: Influence on workflow and efficiency

PONE-D-23-14019R1

Dear Dr. Fistera,

We’re pleased to inform you that your manuscript has been judged scientifically suitable for publication and will be formally accepted for publication once it meets all outstanding technical requirements.

Kind regards,

Vittorio Sambri, M.D., Ph.D.

Academic Editor

PLOS ONE

---

## [Editor Report · Acceptance letter]

26 Jul 2023

PONE-D-23-14019R1 

Point-of-care PCR testing of SARS-CoV-2 in the emergency department: Influence on workflow and efficiency 

Dear Dr. Fistera:

I'm pleased to inform you that your manuscript has been deemed suitable for publication in PLOS ONE. Congratulations! Your manuscript is now with our production department. 

Kind regards, 

on behalf of

Professor Vittorio Sambri 

Academic Editor

PLOS ONE